# Magnetoacoustic Wave Scattering and Dynamic Stress Concentration around the Elliptical Opening in Exponential-Gradient Piezomagnetic Materials

**DOI:** 10.3390/ma15134564

**Published:** 2022-06-29

**Authors:** Zhiwen Wang, Chuanping Zhou, Xueting Zhang, Xiao Han, Junqi Bao, Lingkun Chen, Maofa Wang, Yongping Gong, Weihua Zhou

**Affiliations:** 1School of Mechanical Engineering, Hangzhou Dianzi University, Hangzhou 310018, China; wzw3049972460@hdu.edu.cn (Z.W.); gyp@hdu.edu.cn (Y.G.); 2Hangzhou Changchuan Technology Co., Ltd., Hangzhou 310018, China; qinlongcrcc@foxmail.com (X.H.); baojq@hzcctech.net (J.B.); 3School of Building Science and Engineering, Yangzhou University, Yangzhou 225009, China; lingkunchen08@hotmail.com; 4College of Electrical Engineering, Zhejiang University, Hangzhou 310027, China; dididi@zju.edu.cn

**Keywords:** exponential-gradient piezomagnetic materials, magnetoacoustic coupled dynamics, dynamic stress concentration coefficient, conformal mapping

## Abstract

Based on the theory of magnetoacoustic coupled dynamics, the purpose of this paper is to evaluate the dynamic stress concentration near an elliptical opening in exponential-gradient piezomagnetic materials under the action of antiplane shear waves. By the wave function expansion, the solutions for the acoustic wave fields and magnetic fields can be obtained. Stress analysis is performed by the complex function method and the conformal mapping method, which are used to solve the boundary conditions problem, and is used to express the dynamic stress concentration coefficient (DSCC) theoretically. As cases, numerical results of DSCCs are plotted and discussed with different incident wave numbers and material parameters by numerical simulation. Compared with circular openings, elliptical openings are widely used in material processing techniques and are more difficult to solve. Numerical results show that the dynamic stress concentration coefficient at the elliptical opening is strongly dependent on various parameters, which indicates that the elliptical opening is more likely to cause crack and damage to exponential-gradient piezomagnetic materials.

## 1. Introduction

Functional-gradient piezomagnetic materials, as sophisticated materials in the field of high technology, can adapt to changes in the environment, achieve the mutual transformation of magnetic and mechanical energy, and have unique features of combining sensing, execution, and control. These materials are critical to electronics, radar, microdisplacement control, and aeronautical technologies in the future [1]. Functional-gradient piezomagnetic materials have many advantages over typical component materials; however, they are brittle and have low fracture toughness in nature. The failure mechanism of functional-gradient piezomagnetic materials incorporates flaws such as cavities, fractures, and inclusions under the action of dynamic loadings. The study of its failure mechanism is crucial in the design of devices and components [2].

In recent years, in order to study the structural strength and stability of materials, many scholars have carried out a lot of experimental research and theoretical analysis on the propagation of coupled elastic waves in functional materials, and provided theoretical results at the same time [3,4,5,6,7,8,9]. Hei et al. [3], based on complex function theory, investigated an analytical solution for the dynamic stress concentration due to an arbitrary cylindrical cavity in an infinite inhomogeneous medium. The scattering of arbitrary cavities with variable-coefficient Helmholtz equations is solved by introducing two different conformal maps. Assuming that the density index of the medium changes continuously, the complex-value displacements and stresses of the inhomogeneous medium can be explicitly obtained. With the aid of Green’s function method and the image method, Qi et al. [4] considered the problem of scattering of an SH wave by a semicylindrical salient near-vertical interface in a bimaterial half-space. They calculated the dynamic stress concentration factor around the edge of a semicylindrical salient, and discussed the influences of incident wave number, incident angle, effect of interface and different combinations of material parameters, etc., on the dynamic stress concentration factor. An et al. [5] evaluated the stress concentration at the tip of a permeable interfacial crack near an eccentric elliptical hole in piezoelectric bimaterials under antiplane shearing. Green’s function method and the conformal mapping method, which are used to solve the boundary conditions problem, are used to analyze fracture. In addition, the method proposed can also deal with noneccentric problems and has wider applicability. Based on the study of the dynamic antiplane characteristics for a radial crack emanating from a circular cavity in piezoelectric bimaterials, An et al. [6] established the mechanical model of interfacial cracks emanating from an eccentric circular cavity. Green’s function method, the coordinate transformation method, and conjunction and crack-deviation techniques are adopted to express the dynamic stress intensity factor theoretically. Liu et al. [7] presented the solutions of two-dimensional elastic wave equations in terms of complex wave functions and general expressions for boundary conditions for steady-state incident waves. Dynamic stresses around a cavity of arbitrary shape are then expressed in a series of complex ‘domain functions’; the coefficient of the series can be determined by truncating a set of infinite algebraic equations. Zhao et al. [8] theoretically investigated the problem of dynamically debonded cylindrical inclusion near the interface of semi-infinite piezoelectric materials, and discussed the effects of different geometric and physical parameters on the dynamic stress intensity factor of the crack tip. Sahu et al. [9] studied the propagation behavior of horizontally polarized shear waves in a three-layer composite structure consisting of a piezomagnetic wave layer, a functional-gradient piezoelectric material layer, and an elastic substrate. The effects of layer width, ripple number, and material gradient on horizontally polarized shear waves are obtained.

Research on functional composite materials is also attracting more and more attention. Shin et al. [10] considered the transient response of an interface crack between two dissimilar functionally graded piezoelectric material layers under antiplane shear impact loading, using the integral transform method. The properties of the functionally graded piezoelectric material layers vary continuously along the thickness, and the two layers are connected weak-discontinuously. Jiang et al. [11] researched the dynamic response of a shallow circular inclusion under an incident SH wave in a radially inhomogeneous half-space by applying complex function theory and the multipolar coordinate system. In addition, the mass density of the half-space varies along with the radius direction. Tan et al. [12] presented constitutive equations for the nonlinear electromagnetoelastic properties of piezoelectric/piezomagnetic fiber-reinforced composite materials. These equations are derived on the basis of the thermodynamic principle using Gibbs free energy expanded into a Taylor series, with emphasis being placed on quadratic nonlinearities. Pang et al. [13] analyzed the energy band structure of piezoelectric/piezomagnetic periodic-layered composites and used the transfer matrix method and the stiffness matrix method to calculate the dispersion curve, energy/displacement transmission coefficient, and localization factor to describe the composite material. The band gap is passed, and the corresponding characteristics of the dispersion curve, positioning factor, and response spectrum are obtained. Xue et al. [14] used the wave function expansion method to solve the problem of acoustic wave scattering and dynamic stress concentration around the two openings in e-type piezomagnetic composites. Jiang et al. [15] developed a theoretical method to study the magnetoelastic coupled wave and dynamic stress intensity around a cylindrical aperture in exponential-graded piezomagnetic materials. By employing the decoupling technique, the coupled magnetoelastic governing equations are decomposed, and the numerical examples of the dynamic stress intensity factor near the aperture are presented. The materials used in the above literature include functional-gradient materials, piezomagnetic materials, piezoelectric materials and their composite materials, such as CoFe_2_O_4_/BaTiO_3_ [9], PZT-5H [10], LiNbO_3_/CoFe_2_O_4_ [12], PZT-5H/CoFe_2_O_4_ [12], BaTiO_3_/PZT-4 /CoFe_2_O_4_ [13], and CoFe_2_O_4_ [14].

According to the reviewed literature, there is no suitable method regarding the dynamic antiplane elliptical opening problem of exponential-gradient piezomagnetic materials. Based on the theory of magnetoacoustic coupled dynamics [16,17], the diffraction and dynamic stress concentration around the elliptical opening of exponential-gradient piezomagnetic materials under the action of magnetoacoustic coupled waves have been studied. Exponential-gradient piezomagnetic materials have great prospects in the application of actuation and high-precision sensors, and the study of their dynamic properties can play a strong guiding role in the development of top equipment. The purpose of the present paper is to suggest an effectively theoretical method for solving the dynamic stress concentration coefficient (DSCC) around the elliptical opening in exponential-gradient piezomagnetic materials.

## 2. The Wave Equation and the Solution of the Coupled Field in Exponential-Gradient Piezomagnetic Materials

A cylindrical coordinate system is established in an infinite exponential-gradient piezomagnetic material. The center of the ellipse is the origin, as shown in Figure 1, and the z-direction is the polarization direction. In addition, the material parameters vary exponentially along the polar axis.

The antiplane shear wave propagates along the polar axis. The governing equations expressed by the acoustic field and magnetic potential, in the absence of body forces, can be written as:(1)$$\begin{array}{c} \frac{\tau_{rz}}{r}+\frac{\partial \tau_{rz}}{\partial r}+\frac{1}{r}\frac{\partial \tau_{\theta z}}{\partial \theta }=\rho \frac{\partial^{2}w}{\partial t^{2}}\\ \frac{\;\partial \left(rB_{r}\right)}{\partial r}+\frac{\partial B_{\theta }}{\partial \theta }=0 \end{array}$$
where $\tau_{rz}$ and $\tau_{\theta z}$ are the shear stress components, ρ is the density, w is the displacement in the *z*-direction, and $B_{r}$ and $B_{\theta }$ are the magnetic displacement components.

The constitutive equations of the exponential-gradient piezomagnetic materials can be written as [5]:(2)$$\begin{array}{l} \tau_{rz}=c_{44}\frac{\partial w}{\partial r}+h_{15}\frac{\partial \psi }{\partial r}\\ \tau_{\theta z}=c_{44}\frac{1}{r}\frac{\partial w}{\partial \theta }+h_{15}\frac{1}{r}\frac{\partial \psi }{\partial \theta }\\ B_{r}=h_{15}\frac{\partial w}{\partial r}-\mu_{11}\frac{\partial \psi }{\partial r}\\ B_{\theta }=h_{15}\frac{1}{r}\frac{\partial w}{\partial \theta }-\mu_{11}\frac{1}{r}\frac{\partial \psi }{\partial \theta } \end{array}$$
where $c_{44}$ is the acoustic constants of piezomagnetic materials, $h_{15}$ is the piezomagnetic constant of piezomagnetic materials, $\mu_{11}$ is the magnetic permeability, and ψ is the magnetic potential in materials.

Considering the general situation, the solution of the steady-state wave can be written as:(3)$$w=\tilde{w}e^{-i\omega t}$$
(4)$$\psi =\tilde{\psi }e^{-i\omega t}$$
where ω is the frequency of incident waves, and i is an imaginary unit.

It is assumed that all the material properties of the exponential-gradient piezomagnetic materials have the same exponential function distribution along the polar axis, and the material properties are given as [9]:(5)$$c_{44}=c_{440}e^{2\beta r\cos\theta },\rho =\rho_{0}e^{2\beta r\cos\theta },h_{15}=h_{150}e^{2\beta r\cos\theta },\mu_{11}=\mu_{110}e^{2\beta r\cos\theta }$$
where $c_{440}$ is the initial Young’s modulus, $\rho_{0}$ is the initial density, $h_{150}$ and $\mu_{110}$ are the initial piezomagnetic constant and the initial magnetic permeability, respectively, and β is the inhomogeneity coefficient of the exponential distribution along the polar axis.

Substituting Equation (2) into Equation (1), the expressions are given as:(6)$$\begin{array}{l} \;\;\;c_{440}\left(\nabla^{2}w+2\beta \cos\theta \frac{\partial w}{\partial r}-\frac{2\beta \sin\theta }{r}\frac{\partial w}{\partial \theta }\right)\\ +h_{440}\left(\nabla^{2}\psi +2\beta \cos\theta \frac{\partial \psi }{\partial r}-\frac{2\beta \sin\theta }{r}\frac{\partial \psi }{\partial \theta }\right)=\rho_{0}\frac{\partial^{2}w}{\partial t^{2}}\\ \;\;\;h_{150}\left(\nabla^{2}w+2\beta \cos\theta \frac{\partial w}{\partial r}-\frac{2\beta \sin\theta }{r}\frac{\partial w}{\partial \theta }\right)\\ -\mu_{110}\left(\nabla^{2}\psi +2\beta \cos\theta \frac{\partial \psi }{\partial r}-\frac{2\beta \sin\theta }{r}\frac{\partial \psi }{\partial \theta }\right)=0 \end{array}$$
where $\nabla^{2}=\frac{\partial^{2}}{\partial r^{2}}+\frac{1}{r}\frac{\partial }{\partial r}+\frac{1}{r^{2}}\frac{\partial^{2}}{\partial \theta^{2}}$ is the polar coordinate Laplacian operator.

To simplify the calculation by introducing a new function $\varphi =\psi -\left(h_{150}/\mu_{110}\right)w$, Equation (6) can be simplified as:(7)$$\nabla^{2}w+2\beta \cos\theta \frac{\partial w}{\partial r}-\frac{2\beta \sin\theta }{r}\frac{\partial w}{\partial \theta }=\xi^{2}\frac{\partial^{2}w}{\partial t^{2}}$$
(8)$$\nabla^{2}\varphi +2\beta \cos\theta \frac{\partial \varphi }{\partial r}-\frac{2\beta \sin\theta }{r}\frac{\partial \varphi }{\partial \theta }=0$$
where $\xi =\sqrt{\rho_{0}/\chi }$ is the reciprocal of the propagation velocity of the antiplane shear waves, and $\chi =c_{440}+h_{150}^{2}/\mu_{110}$.

The steady-state solution to this problem is studied. Let $w=w_{0}We^{-i\omega t}$, and Equation (7) can be written as:(9)$$\nabla^{2}W+2\beta \left(\cos\theta \frac{\partial W}{\partial r}-\frac{\sin\theta }{r}\frac{\partial W}{\partial \theta }\right)+k^{2}W=0$$
where ω is the incident wave frequency, and k=ωξ is the incident wave number. 

We can write the conformational solution of Equation (9) as:(10)$$W=e^{-\beta r\cos\theta }f\left(r,\theta \right)$$
where f(r,θ) is a constructor.

Substituting Equation (10) into Equation (9) yields the standard Helmholtz equation:(11)$$\nabla^{2}f+\kappa^{2}f=0$$
where $\kappa =\sqrt{k^{2}-\beta^{2}}$.

Denote $\chi_{1}=h_{150}/\mu_{110}$, we also can derive the magnetic potential as:(12)$$\varphi =w_{0}\chi_{1}e^{-\beta r\cos\theta }e^{-i\left(i\beta r\cos\theta -\omega t\right)}$$

Based on the complex function theory, the complex variable can be defined as $z=r\left(\theta \right)e^{i\theta }$. Because of the time-harmonic behavior of all field quantities, the common multiplier $e^{-i\omega t}$ is suppressed here and in the following [13]. We use an infinite series to express the incident wave, and the expression for incident waves stratifying can be expressed as [8]:
(13)w(i)=w0e−βRe(z)∑n=−∞∞inJn(κ|z|){z|z|}n
(14)φ(i)=w0χ1e−βRe(z)∑n=−∞∞inJn(iβ|z|){z|z|}n

Similarly, the form of the scattering waves in Equations (7) and (8) can be written as [8]:(15)w(s)=w0e−βRe(z)∑n=−∞∞AnHn(1)(κ|z|){z|z|}n
(16)φ(s)=w0χ1e−βRe(z)∑n=−∞∞BnHn(1)(iβ|z|){z|z|}n
where $A_{n}$ is the undetermined coefficient to describe the scattered acoustic field, $B_{n}$ is the undetermined coefficient to describe the scattered magnetic field, and $H_{n}^{\left(1\right)}(\cdot )$ is the *n*th order Bessel function of the third kind.

Taking the incident field and scattered field together, the total field of acoustic in the exponential-gradient piezomagnetic materials is expressed as:(17)$$w^{\left(t\right)}=w^{\left(i\right)}+w^{\left(s\right)}$$

The total magnetic potential is expressed as:(18)$$\psi^{\left(t\right)}=\varphi^{\left(i\right)}+\varphi^{\left(s\right)}+\chi_{1}w^{\left(t\right)}$$

Without loss of generality, it should be noted that there is only a magnetic field in the opening, which can be expressed as:(19)ψc=w0χ1e−βRe(z)∑n=−∞∞CnJn(iβ|z|){z|z|}n
where $C_{n}$ is the undetermined coefficient describing the internal magnetic field.

## 3. Determine the Boundary Conditions and Modal Coefficients of the Elliptical Opening

Considering the case of a noncircular opening with a smooth boundary, the method of conformal mapping is used, and the complex variables can be defined as λ=ε+iη and $\bar{\lambda }=\varepsilon -i\eta$. Therefore, the conformal mapping function can be taken as [2]:(20)$$z=\Omega \left(\lambda \right)=R\left(\lambda +\sum_{l=1}^{\infty }C_{l}\lambda^{-l}\right)$$
where R is the real constant, and $C_{l}$ is the complex constant.

The expression of the elliptical opening under the conformal mapping function is:(21)$$z=\Omega \left(\lambda \right)=\frac{a}{1+m}\left(\lambda +\frac{m}{\lambda }\right),\lambda =\zeta e^{i\alpha }$$
where m∈(0,1) is the relative eccentricity of the elliptical opening.

At the opening interface of an infinite exponential-gradient piezomagnetic materials, the boundary conditions of free stress and magnetic continuous are as follows:(22)$$\left\{ \begin{array}{l} {\tau_{rz}|}_{r=\left|z\right|}=0\\ {B_{r}|}_{r=\left|z\right|}={B_{r}^{c}|}_{r=\left|z\right|}\\ {\psi^{\left(t\right)}|}_{r=\left|z\right|}={\psi^{c}|}_{r=\left|z\right|} \end{array}\right.$$

Substituting Equations (2), (18), and (19) into Equation (22), by magnetoacoustic coupled theory, a definite infinite system of linear algebraic equations can be obtained [3]:(23)$$\sum_{n=-\infty }^{\infty }E_{n}X_{n}=E_{i},(i=1,2,3)$$
where
(24)$${Ε}_{n}=\left[\begin{array}{ccc} E_{11}^{} & E_{12}^{} & E_{13}^{}\\ E_{21}^{} & E_{22}^{} & E_{23}^{}\\ E_{31}^{} & E_{32}^{} & E_{33}^{} \end{array}\right]X_{n}^{}=\left[\begin{array}{c} A_{n}\\ B_{n}\\ C_{n} \end{array}\right]Ε=\left[\begin{array}{c} E_{1}\\ E_{2}\\ E_{3} \end{array}\right]$$

Multiply both sides of Equation (23) by $e^{-is\theta }$ and integrate on the interval (−π,π), and Equation (23) can be transformed into:(25)$$\sum_{n=-\infty }^{\infty }E_{ns}X_{n}=E_{s}$$
where $E_{ns}=\frac{1}{2\pi }\int_{-\pi }^{\pi }E_{n}e^{-is\theta }d\theta$, $E_{s}=\frac{1}{2\pi }\int_{-\pi }^{\pi }E_{i}e^{-is\theta }d\theta$.

Using Equation (25), the mode coefficients $A_{n}$, $B_{n}$, and $C_{n}$ can be obtained, where n=−∞∼+∞.

## 4. Dynamic Stress Concentration Coefficient

In the complex plane, Equation (2) can be rewritten as:(26)τθz=c150e2βRe(z)(∂wt∂zz|z|+∂wt∂z¯z¯|z|)+h150e2βRe(z)(∂ψt∂zz|z|+∂ψt∂z¯z¯|z|)

Substituting Equations (17) and (18) into Equation (26), we can obtain:(27)τθz=1κχkw0rΩ′(λ)e(−β+iκ)Re(Ω(λ))(β−iκ)Im[λΩ′(λ)]+1κw01rΩ′(λ)e−βRe(Ω(λ))×∑n=−∞∞An{[βIm(λΩ′(λ))+inRe(λΩ′(λ)Ω(λ))]Hn(1)(κ|Ω(λ)|)−κ2Im(λΩ(λ)¯|Ω(λ)|Ω′(λ))[Hn−1(1)(κ|Ω(λ)|)−Hn+1(1)(κ|Ω(λ)|)]}{Ω(λ)|Ω(λ)|}n
and Equation (27) uses the following equations [7].
(28)$$\begin{array}{l} \frac{\partial }{\partial z}\left[H_{n}\left(\kappa \left|z\right|\right){\left\{\frac{z}{\left|z\right|}\right\}}^{n}\right]=\frac{\kappa }{2}H_{n-1}\left(\kappa \left|z\right|\right){\left\{\frac{z}{\left|z\right|}\right\}}^{n-1}z^{\prime }\\ \frac{\partial }{\partial \bar{z}}\left[H_{n}\left(\kappa \left|z\right|\right){\left\{\frac{z}{\left|z\right|}\right\}}^{n}\right]=-\frac{\kappa }{2}H_{n+1}(\kappa \left|z\right|){\left\{\frac{z}{\left|z\right|}\right\}}^{n+1}\bar{z^{\prime }} \end{array}$$
where z=Ω(λ).

According to the definition of *DSCC*, which is the ratio of the circumferential shear stress in the opening to the stress amplitude [4]:(29)$$DSCC=\left|\tau_{\theta z}/\tau_{0}\right|$$
where $\tau_{\theta z}$ is dimensionless stress, which represents dynamic stress concentration, and $\tau_{0}=\chi kw_{0}$ is the amplitude of the incident wave.

Therefore, from Equation (29), we can obtain that the DSCC around the elliptical opening in the exponential-gradient piezomagnetic materials is expressed as:(30)DSCC=|1κ1rΩ′(λ)e(−β+iκ)Re(Ω(λ))(β−iκ)Im[λΩ′(λ)]+1κw01rΩ′(λ)e−βRe(Ω(λ))×∑n=−∞∞An{[βIm(λΩ′(λ))+inRe(λΩ′(λ)Ω(λ))]Hn(1)(κ|Ω(λ)|)−κ2Im(λΩ(λ)¯|Ω(λ)|Ω′(λ))[Hn−1(1)(κ|Ω(λ)|)−Hn+1(1)(κ|Ω(λ)|)]}{Ω(λ)|Ω(λ)|}n|

## 5. Numerical Examples and Discussion

Analysis of the above expressions reveals Equation (30) is a convergent infinite-series equation; when n≥15, the result satisfies the engineering accuracy. Simultaneously, we choose NiOFe_2_O_3_ as the example of the exponential-gradient piezomagnetic materials. The related material properties are:(31)$$\begin{array}{c} \rho_{0}=5.35\times 10^{3}\mathrm{kg}\cdot m^{-2},c_{440}=1.7\times 10^{10}N\cdot m^{-2}\\ h_{150}=454N\cdot A^{-1}\cdot m^{-1},\mu_{110}=5.53\times 10^{-5}N\cdot A^{-2} \end{array}$$

In the numerical results, the variables are dimensionless. The dimensionless incident wave numbers ka∈(0.1,6) are taken, and βa∈(−0.1,0.1) are the material parameters.

It can be obtained from the data in Figure 2, Figure 3, Figure 4, Figure 5, Figure 6 and Figure 7 how the DSCC is distributed around the opening at different wave numbers ka and various relative eccentricity m. The dimensionless incident wave numbers ka=0.1,1.0,2.0, the relative eccentricities m=1/6,1/4, and the material parameters βa=−0.05,0,0.025,0.05. According to the results from the figures, it seems that the DSCC distribution in the opening is strongly influenced by the relative eccentricity m and the material parameters βa.

Taking ka=0.1 and βa=−0.05,0,0.025,0.05, the stress amplitudes around the elliptical opening are 2.40, 2.15, 2.08, and 2.41, as shown in Figure 2 (taking m=1/6), but change to 3.85, 3.46, 3.36, and 3.87 in Figure 3 (taking m=1/4). Interestingly, the data between Figure 2 and Figure 3 show that the big relative eccentricity would make the DSCC amplitude change easily.

Taking the discussion further, it can be observed that the DSCC distribution in the opening in Figure 2 and Figure 3 has an obvious pattern of ka=0.1: the distribution of the DSCC in the opening is symmetric about the vertical and horizontal axes, and its maximum value occurs at θ=π/2 and θ=3π/2, which meets the engineering experience.

With the frequency of the incident wave increasing, as shown in Figure 4 (ka=1.0) and Figure 5 (ka=1.0), the antiplane shear wave has a strong effect on the distribution of DSCC. Comparing Figure 2 and Figure 4, two changes in DSCC distribution can be obtained: the first one is the DSCC distribution on the oncoming side is smaller than the back distribution, and then the values of the DSCC, which are located in the opening at high frequency, are significantly lower than those at low frequency, which indicates that the incident wave can change the material stiffness to some extent.

As seen in Figure 6 and Figure 7, the main lobe position is shifted to the oncoming side differently from the low-frequency case. The DSCC on the oncoming side is larger than that at the back side. The side lobes increase, and the whole distribution of the DSCC remains symmetric about the horizontal axis.

Figure 8 shows the trend of the DSCC amplitude in the opening with the incident wave number ka (m=1/6). From the figure, it can be found that the stress amplitude has a specific undulating pattern, and the effect of the material parameter βa on the DSCC amplitude changes as ka changes.

According to the above discussion, the DSCC has a great correlation with wave number ka. Reasonable design parameters for the environment can greatly increase the service life of materials. In practice, most materials operate at low frequencies, so small openings need to be considered. A material with high-frequency vibration can increase the material parameter βa appropriately to obtain better material stiffness.

## 6. Conclusions

The purpose of this paper is to research the response of stress concentrations to incident wave numbers and material parameters in exponential-gradient piezomagnetic materials. The expression for the considered antiplane shear wave has been obtained in a complex function. The material properties have an exponential variation, and the relative eccentricity is variable.

To deal with the complexity of elliptical openings, conformal mapping is used to map the ellipse to the unit circle for the theoretical calculation. The displacement and stress components are represented by the superposition of wave functions, which are the Bessel and Hankel functions. The modal coefficients can be calculated by the boundary conditions at the opening and a nonlinear system of equations. One of the highlights of this paper is that the problem is transformed from the real number field to the complex number field by the complex function method, which greatly simplifies the complexity of the problem. In addition, the nonlinear system of equations is transformed into the solvable linear system of equations by orthogonalization. This strategy provides a solution to similar challenges.

More details can be obtained from numerical calculations:The relative eccentricity is significant to the DSCC among the material parameters. It is important to reduce relative eccentricity when designing elliptical openings in the exponential-gradient piezomagnetic materials.With other parameters unchanged, the dynamic stress concentration coefficient increases with the increase in relative eccentricity. We can regard that as the relative eccentricity increases, the material stiffness decreases.As the incident wave number ka
grows, the DSCC amplitude rises and falls significantly. The DSCC amplitude is effectively suppressed by the incident wave number ka in a specific frequency range. This property can provide reference for engineering processing.
The material parameter βa
of exponential-gradient piezomagnetic materials significantly impacts the amplitude of the DSCC with the elliptical opening. The material parameter βa is inversely proportional to the amplitude of the DSCC under a larger incident wave number (ka>4.5).


## Figures and Tables

**Figure 1 materials-15-04564-f001:**
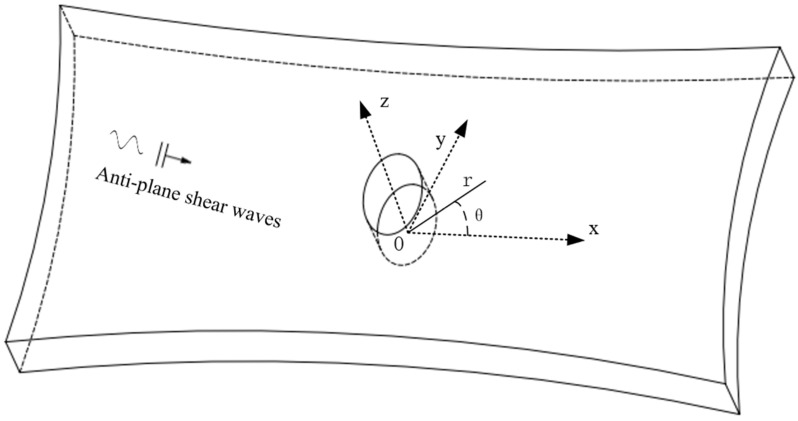
Antiplane shear waves are incident into an elliptical opening of an exponential-gradient piezomagnetic material.

**Figure 2 materials-15-04564-f002:**
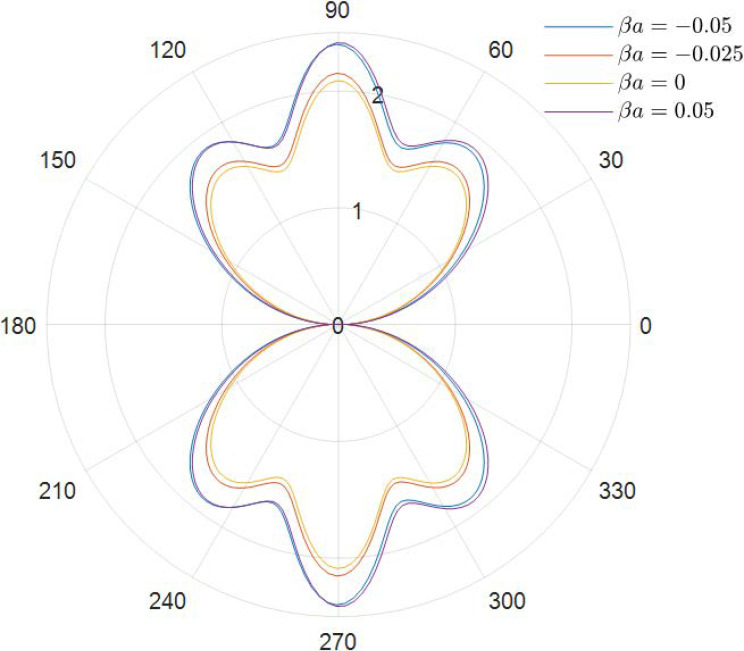
Polar graph of DSCC around the elliptical opening with *ka* = 0.1 and *m* = 1/6.

**Figure 3 materials-15-04564-f003:**
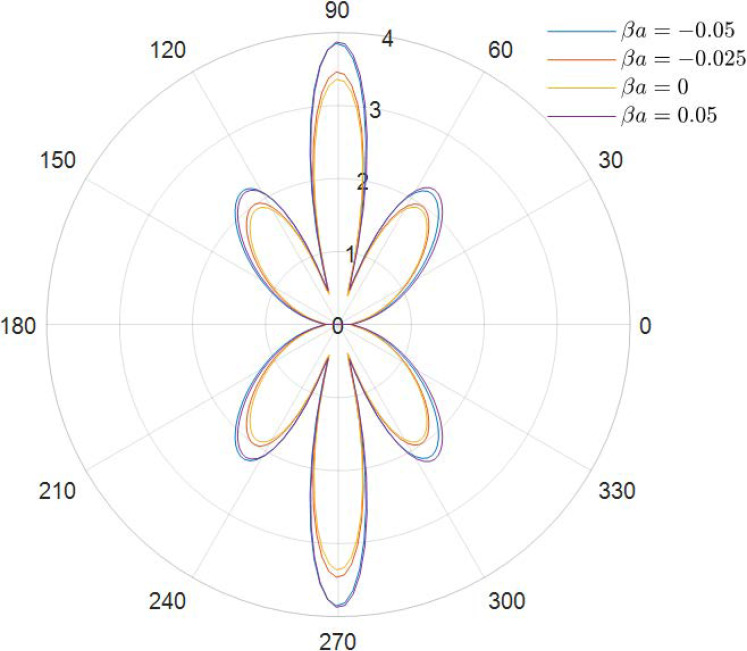
Polar graph of DSCC around the elliptical opening with *ka* = 0.1 and *m* = 1/4.

**Figure 4 materials-15-04564-f004:**
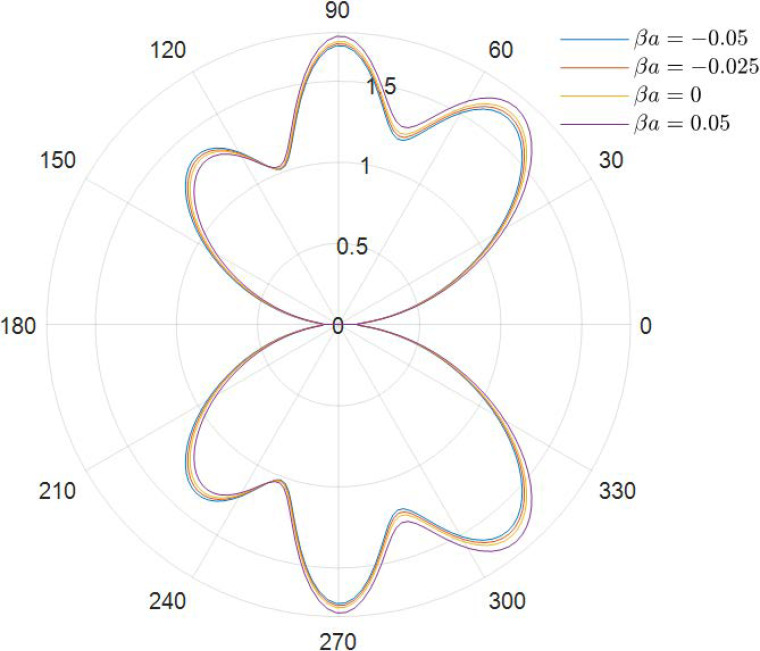
Polar graph of DSCC around the elliptical opening with *ka* = 1.0 and *m* = 1/6.

**Figure 5 materials-15-04564-f005:**
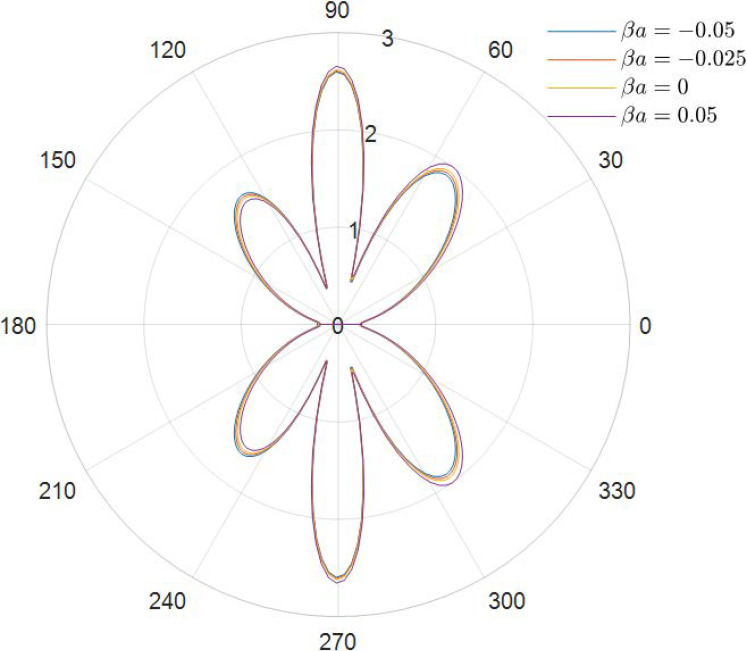
Polar graph of DSCC around the elliptical opening with *ka* = 1.0 and *m* = 1/4.

**Figure 6 materials-15-04564-f006:**
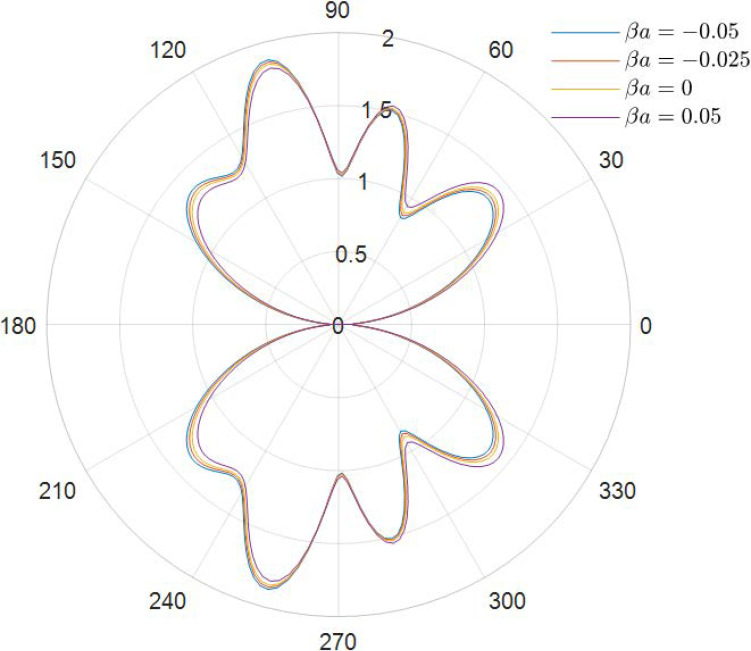
Polar graph of DSCF around the elliptical opening with *ka* = 2.0 and *m* = 1/6.

**Figure 7 materials-15-04564-f007:**
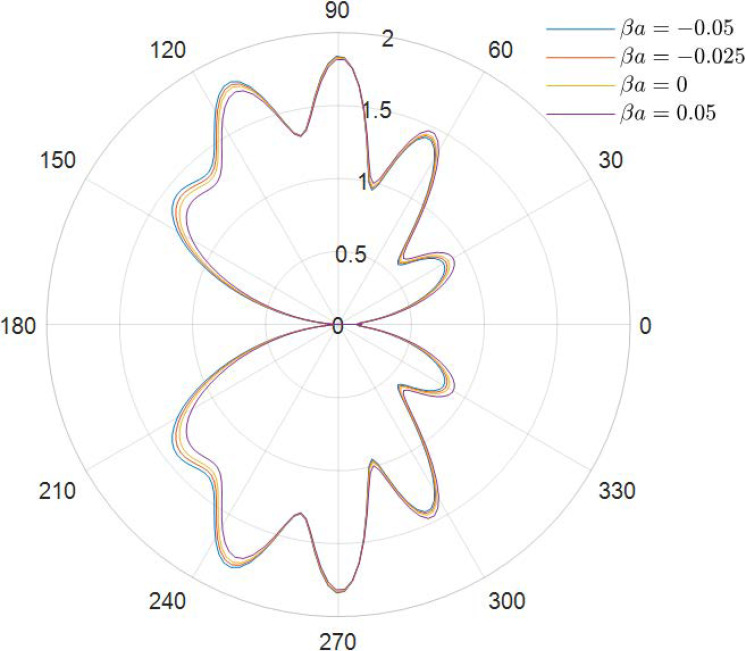
Polar graph of DSCF around the elliptical opening with *ka* = 4.0 and *m* = 1/6.

**Figure 8 materials-15-04564-f008:**
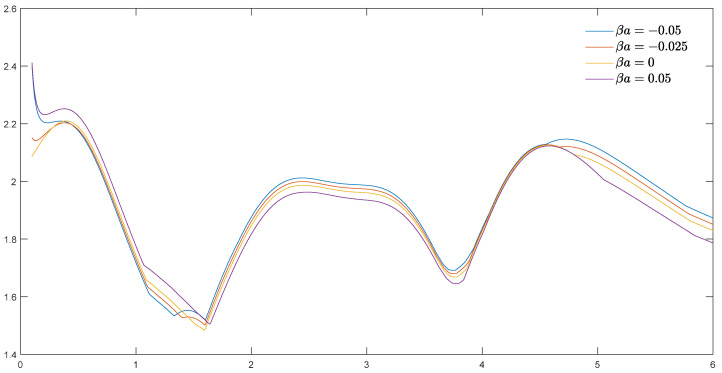
Maximum DSCC and incident wave number *ka* around elliptical opening with *m* = 1/6.

## Data Availability

Data sharing is not applicable for this article.

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
