# Peer review of "Magnetoacoustic Wave Scattering and Dynamic Stress Concentration around the Elliptical Opening in Exponential-Gradient Piezomagnetic Materials"

_materials, 2022, doi:10.3390/ma15134564_

Round 1

Reviewer 1 Report

1. In my previous review of this paper, I proposed that the authors take the help of a native English speaker to clean up the technical issues. Unfortunately, the authors did not take any effort to address this major concern from this reviewer. The said they took help of a native speaker. However, Just reading the first two lines of the manuscript (as below), the statements don't make much sense and there are technical writing issues. 

"Functional gradient material is a new type of functional material whose properties and functions vary gradient. with the interface removing, properties of materials vary with the change in space smoothly"

The manuscript needs to be strictly checked by a native speaker, and statements need to be written carefully so that they make sense.

2. The authors did not take into account the reflected wave-fields, mentioning it is an infinite medium. They only consider incident and scattered wave-fields. This seems very inappropriate to the reviewer, as it is completely impractical to make such assumption.  In reality, it is bound to have considering the reflected wave fields to gain the physical insight and achieve relevance with reality. You may make further assumptions to alleviate the complexity of equations if required.

3. The authors did not take into account the non-linear elasticity. I see major portion of work done in linear regime (in literature too), which makes the novelty of current work below average.

The state-of-the-art has been improved, though. Also, some other aspects/concerns were addressed in this round. Please focus on the major concerns as mentioned above.

Unfortunately, in this reviewers' opinion, the current version still lacks the required quality to be able to be published in materials.

Reviewer 2 Report

This research work Magneto-acoustic wave scattering and dynamic stress 2 concentration around the elliptical opening in the exponential 3 gradient piezomagnetic materials, the theoretical examine materials condition and analyze the with different parameters. 

The objective or motive of the present work is not well elaborated in the work, author have take care to explore motive of work in introduction as well have to describe major innovation of work in conclusion session.

Reviewer 3 Report

Manuscript Number materials-1752277

Title: Magneto-acoustic wave scattering and dynamic stress concentration around the elliptical opening in the exponential gradient piezomagnetic materials

Journal: materials

Recommendation:

minor revision 

Overview:

Theoretical study of magneto-acoustic wave scattering and dynamic stress concentration around the elliptical opening in the exponential gradient piezomagnetic materials seems very attractive in relation to magnetoelectric (ME) materials.

However, I found some inaccuracies in the manuscript. Therefore, I recommend a minor revision. My questions and comments are detailed below.

11     Please add information to the introduction about magnetostrictive materials for which that theory can be useful.

22     It will be interesting to add discussion how prediction of presented theory can improve ME composite materials and increase ME coefficient.

33    Where can we find exponential gradient piezomagnetic materials?

Round 2

Reviewer 1 Report

The paper can be accepted in current form.